# Meta Learning the Step Size in Policy Gradient Methods

**Luca Sabbioni**                                                   luca.sabbioni@polimi.it
**Francesco Corda**                                     francesco.corda@mail.polimi.it
**Marcello Restelli**                                      marcello.restelli@poimi.it
*Department of Electronics, Information and Bioengineering, Politecnico di Milano, Italy*

## Abstract

Policy-based algorithms are among the most widely adopted techniques in model-free RL, thanks to their strong theoretical groundings and good properties in continuous action spaces. Unfortunately, these methods require precise and problem-specific hyperparameter tuning to achieve good performance and, as a consequence, they tend to struggle when asked to accomplish a series of heterogeneous tasks. In particular, the selection of the step size has a crucial impact on the ability to learn a highly performing policy, affecting the speed and the stability of the training process, and often being the main culprit for poor results. In this paper, we tackle these issues with a Meta Reinforcement Learning approach, by introducing a new formulation, known as meta-MDP, that can be used to solve any hyperparameter selection problem in RL with contextual processes. After providing a theoretical Lipschitz bound to the performance in different tasks, we adopt the proposed framework to train a batch RL algorithm to dynamically recommend the most adequate step size for different policies and tasks. In conclusion, we present an experimental campaign to show the advantages of selecting an adaptive learning rate in heterogeneous environments.

**Keywords:** Meta Reinforcement Learning, Hyperparameter tuning, Policy Gradient

## 1. Introduction

The main goal of reinforcement learning (RL, Sutton and Barto 1998) is to build an agent capable of learning a behavior that maximizes the amount of reward collected while interacting with the environment. Typically, these environments are modelled as Markov decision processes (MDP, Puterman 2014), where all trajectories share the same underlying distribution. Nevertheless, in many real world scenarios there can be exogenous variables affecting the whole process; one might think for example of a car race, where the road temperature, or the tyre choice may require different strategies. One of the most successful stream of model-free RL applications adopts policy based algorithms, that provide strong theoretical groundings and good empirical properties in continuous action spaces. Unfortunately, these methods require precise and problem-specific hyperparameter tuning to achieve good performance, causing them to struggle when applied on a series of heterogeneous tasks. The fundamental parameter to tune is the stepsize, which has a crucial impact on the ability to learn a highly performing policy, affecting the speed and the stability of the training process, and often being the main responsible for poor results. Hyperparameter optimization is an important component of automated machine learning (Bergstra and Bengio, 2012; Snoek et al., 2012; Meier et al., 2018; Zhu et al., 2019). However, these works are seldom adopted in RL (as Paul et al. 2019; Xu et al. 2018; Paine et al. 2020) as they become computationally intractable and sample inefficient. In this work, we consider the specific problem of learning how to dynamically select the best step size for each policy; moreover, we consider the case

where the MDP process might differ due to the presence of the exogenous variable, here denoted as "tasks" or "contexts". To start,we propose a formalization of the problem by defining a meta-MDP.This general framework allows to solve a set of RL tasks, grouped as a contextual Markov decision process (Hallak et al., 2015). We discuss the main elements of the model, such as the meta objective function, which measures the return variations, and the meta action, consisting in the hyperparameter selection for a policy update. Then we consider Lipschitz meta-MDPs, in which the trajectories sampled from similar contexts have comparable properties, and we derive some guarantees on the expected return. Subsequently, we propose to learn the step size of policy gradient methods in a meta-MDP. We accomplish this through the application of a value based algorithm, known as Fitted Q-Iteration (FQI), used to dynamically recommend the most adequate step size in the current scenario. In conclusion, we evaluate our approach in various simulated environments.

## 2. Preliminaries

A discrete-time MDP is defined as a tuple $\langle \mathcal{S}, \mathcal{A}, \mathcal{P}, \mathcal{R}, \gamma, \mu \rangle$, where $\mathcal{S}$ and $\mathcal{A}$ are the state space and the action space, $\mathcal{P}(\cdot|s,a)$ is the Markovian transition, which assigns to each state-action pair $(s,a)$ the probability of reaching the next state $s'$, $\mathcal{R}$ is the reward distribution, with expected value $r(\cdot|s,a)$, bounded by hypothesis, i.e. $\sup_{s \in \mathcal{S}, a \in \mathcal{A}} |\mathcal{R}(s,a)| \le R_{\max}$. Finally, $\gamma \in [0,1]$ is the discount factor, and $\mu$ is the distribution of the initial state. The policy of an agent, denoted as $\pi(\cdot|s)$, assigns to each state $s$ a density distribution over the action space $\mathcal{A}$. We can define the return of a trajectory $\tau := (s_0, a_0, s_1, a_1, s_2, a_2, ..., a_{H-1}, s_H)$ with horizon $H$ as the discounted sum of the reward collected: $G_\tau = \sum_{t=0}^{H} \gamma^t \mathcal{R}(s_t, a_t)$. Consequently, it is possible to define the expected return $j_\pi$ as the expected performance under policy $\pi$. In an analogous way, we can define the value function $V_\pi(s)$ and the action-value function $Q_\pi(s,a)$ as the expected return obtained by starting respectively from the state $s$ or from the pair $(s,a)$ and then following the policy $\pi$. For the rest of the paper, we consider parametric policies, where the policy $\pi_{\boldsymbol{\theta}}$ is parametrized by a vector $\boldsymbol{\theta} \in \Theta \subseteq \mathbb{R}^m$. In this case, the goal is to find the optimal parametric policy maximizing the performance, i.e. $\boldsymbol{\theta}^* = \arg\max_{\boldsymbol{\theta} \in \Theta} j(\boldsymbol{\theta})$. Policy-based algorithms adopt a gradient-ascent approach, where the policy gradient $\widehat{\nabla}_N j(\boldsymbol{\theta})$ can be estimated from a batch of $N$ trajectories (Sutton et al., 2000). An important variation on the approach consists in following the steepest ascent direction by means of the natural policy gradient (Kakade, 2001), which includes information regarding the curvature of the return manifold over the policy space in the form of the Fisher information matrix. We denote its estimator as $\widehat{g}_N(\boldsymbol{\theta})$.

**Lipschitz MDP.** Let $(\mathcal{X}, d_{\mathcal{X}})$, $(\mathcal{Y}, d_{\mathcal{Y}})$ be two metric spaces; a function $f : \mathcal{X} \to \mathcal{Y}$ is called $L_f$-Lipschitz continuous ($L_f$-LC), if $d_{\mathcal{Y}}(f(x), f(x')) \le L_f d_{\mathcal{X}}(x, x') \forall x, x' \in \mathcal{X}$. Moreover, we define the Lipschitz semi-norm as $\|f\|_L = \sup_{x,x' \in \mathcal{X}: x \ne x'} \frac{d_{\mathcal{Y}}(f(x), f(x'))}{d_{\mathcal{X}}(x, x')}$. For real functions, the usual metric is the Euclidean distance while, for distributions, a common metric is the Kantorovich, or $L^1$-Wasserstein distance $\mathcal{K}$. Rachelson and Lagoudakis (2010b); Pirotta et al. (2015) introduced some notion of smoothness in RL by defining the Lipschitz-MDP. Let $\mathcal{M}$ be an MDP, $\mathcal{M}$ is called $(L_P, L_r)$-LC if for all $(s,a), (\overline{s}, \overline{a}) \in \mathcal{S} \times \mathcal{A}$:

$$\mathcal{K}\left(P(\cdot|s,a), P(\cdot|\overline{s},\overline{a})\right) \le L_P \, d_{\mathcal{S} \times \mathcal{A}}\left((s,a), (\overline{s},\overline{a})\right), \quad |r(s,a) - r(\overline{s},\overline{a})| \le L_r \, d_{\mathcal{S} \times \mathcal{A}}\left((s,a), (\overline{s},\overline{a})\right).$$

Moreover, a policy $\pi$ is called $L_\pi$-LC if $\mathcal{K}\left(\pi(\cdot|s), \pi(\cdot|\overline{s})\right) \leq L_\pi\, d_\mathcal{S}\left(s, \overline{s}\right) \forall s, \overline{s} \in \mathcal{S}$. An important result is that, for a $(L_p, L_r) - LC$ MDP, and a $L_\pi - LC$ policy, also the expected return, the $Q-$function and the gradient components are Lipschitz continuous w.r.t. $\boldsymbol{\theta}$.

**Meta Reinforcement Learning.** As suggested by the name, meta learning implies a higher level of abstraction with respect to regular machine learning. In particular, meta reinforcement learning (meta-RL) consists in applying meta learning techniques to RL tasks. Usually, these tasks are formalized in MDPs by a common set of parameters, known as the *context* $\omega$. The natural candidate to represent the set of RL tasks is the contextual Markov decision process (CMDP), defined by Hallak et al. (2015) as a tuple $(\Omega, \mathcal{S}, \mathcal{A}, \mathcal{M}(\omega))$ where $\Omega$ is called the context space, $\mathcal{S}$ and $\mathcal{A}$ are the shared state and action space, and $\mathcal{M}$ is function mapping any context $\omega \in \Omega$ to an MDP, such that $\mathcal{M}(\omega) = \langle \mathcal{S}, \mathcal{A}, P_\omega, R_\omega, \gamma_\omega, \mu_\omega \rangle$. In other words, a CMDP includes in a single entity a group of tasks that share the same state and action space. In the following sections, we will also assume that $\gamma$ and $\mu$ are shared across all the tasks.

## 3. Meta-MDP

We now present the concept of meta-MDP, a framework to solve meta-RL tasks that extends the CMDP model by including the learning model and the policy parametrization. Similar approaches to the one proposed in this section can be found in Garcia and Thomas (2019). To start, let's consider the various tasks used in a meta-training procedure as a set of MDPs such that each task $\mathcal{M}_\omega$ can be sampled from the distribution $\psi$ defined on the context space $\Omega$. This set can be equivalently seen as a CMDP $\mathscr{M} = \langle \Omega, \mathcal{S}, \mathcal{A}, \mathcal{M}(\omega) \rangle$, where $\mathcal{M}(\omega) = \mathcal{M}_\omega$. Similarly, we define a distribution $\rho$ over the policy space $\boldsymbol{\Theta}$, in such a way that at each iteration in an MDP $\mathcal{M}_\omega$, the policy parameters $\boldsymbol{\theta}_0$ are initialized to a value sampled from $\rho$. We assume to be able to explicitly represent the task $\omega$.

**Definition 3.1** *A meta-MDP is a tuple $\langle \mathcal{X}, \mathcal{H}, \mathcal{L}, \widetilde{\gamma}, (\mathscr{M}, \psi), (\boldsymbol{\theta}, \rho), f \rangle$, where:*

- *$\mathcal{X}$ and $\mathcal{H}$ are respectively the meta observation space and the learning action;*
- *$\mathcal{L} : \boldsymbol{\Theta} \times \Omega \times \mathcal{H} \to \mathbb{R}$ is the meta reward function;*
- *$\widetilde{\gamma}$ is the meta-discount factor;*
- *$(\mathscr{M}, \psi)$ and $(\boldsymbol{\theta}, \rho)$ contain respectively a CMDP $\mathscr{M}$ with distribution over tasks $\psi$, and the policy space, with initial distribution $\rho$;*
- *$f : \boldsymbol{\Theta} \times \mathcal{H} \to \boldsymbol{\Theta}$ is the update rule of the learning model chosen.*

In particular, a meta-MDP attempts to enclose the general elements necessary to learn a RL task into a model with similar properties to a classic MDP. The meta observation space $\mathcal{X}$ of a meta-MDP can be considered as the generalization of the observation space in classic MDPs. In order to have a complete information about the current position, it is necessary to include in the state the policy $\boldsymbol{\theta}_t$ and the task $\omega$; moreover, relevant information of the overall process is included in the gradient estimation. Consequently, in the following we will model the meta state as the concatenation of all these features.

Each action $h \in \mathcal{H}$ performed on the meta-MDP determines a specific hyperparameter that regulates the update rule $f$, i.e. $\boldsymbol{\theta}_{k+1} = f\left(\boldsymbol{\theta}_k, h_k\right)$. In particular, in this work we focus

on (normalized) natural gradient ascent (NGA), in which the action $h$ determines the step size. Similarly to a standard RL problem, the training of a meta-MDP is accomplished by optimizing a reward function. Meta-learning has the main goal of learning to learn: as a consequence we want to consider learning as our reward, hence we define $\mathcal{L}(\boldsymbol{\theta}, \omega, h)$ as the performance gain obtained after one update of the policy $\boldsymbol{\theta}$ with action $h$ under context $\omega$: $\mathcal{L}(\boldsymbol{\theta}, \omega, h) := j_\omega(f(\boldsymbol{\theta}, h)) - j_\omega(\boldsymbol{\theta})$.

Differently from a standard MDP, a meta-MDP does not include a Markovian transition model that regulates its dynamics: given $x_t$, the transition to the next state $x_{k+1}$ is, of course, stochastic, but it implicitly depends only on the distribution of the trajectories induced by the pair $(\theta_k, \mathcal{M}_\omega)$ and on the update rule $f$. The initial state hence depends on $\psi$ and $\rho$. The choice of the meta-discount factor $\widetilde{\gamma}$ is critical: meta-learning is very often considered as paired with few-shot learning, where a short horizon is taken into account for the learning process. However, a myopic behavior induced by a low discount factor might lead to prefer actions leading to local optima, while it might be necessary to take more cautious steps in order to get to the global optima of the learning process. For this reason, we set $\widetilde{\gamma} = 1$.

## 4. Context Lipschitz Continuity

In this section, we consider a meta-MDP in which all inner tasks satisfy the Lipschitz continuity assumption. Under this condition, we are able to derive a bound on the approximation errors obtained by the meta-agent when acting on unseen tasks. Let's suppose to be provided with a CMDP, where all the inner MDPs are Lipschitz. Moreover, let's also assume that the set of MDPs is Lipschitz continuous w.r.t the context $\omega$:

**Assumption 4.1** *Let $\mathcal{M}$ be a CMDP. $\mathcal{M}$ is called $(L_{\omega_P}, L_{\omega_r})$-Context Lipschitz Continuous $((L_{\omega_P}, L_{\omega_r})$-CLC) if for all $(s, a), (\overline{s}, \overline{a}) \in \mathcal{S} \times \mathcal{A}$, $\forall \omega, \widehat{\omega} \in \Omega$:*

$$\mathcal{K}\left(P_\omega(\cdot \mid s, a), P_{\widehat{\omega}}(\cdot \mid s, a))\right) \le L_{\omega_P} d_\Omega(\omega, \widehat{\omega}) \qquad \left|R_\omega(s, a) - R_{\widehat{\omega}}(s, a)\right| \le L_{\omega_r} d_\Omega(\omega, \widehat{\omega});$$

This means that we have some notion of continuity w.r.t. the task: when two MDPs with similar contexts are considered, then their transition and reward processes are similar.

**Theorem 4.1** *Let $\mathcal{M}$ be a $(L_{\omega_P}, L_{\omega_r})$-CLC CMDP for which $\mathcal{M}(\omega)$ is $(L_P(\omega), L_r(\omega))$-LC $\forall \omega \in \Omega$. Given a $L_\pi$-LC policy $\pi$, the action value function $Q_\omega^\pi(s, a)$ is $L_{\omega_Q}$-CLC w.r.t. the context $\omega$, i.e.:*

$$\left|Q_\omega^\pi(s, a) - Q_{\widehat{\omega}}^\pi(s, a)\right| \le L_{\omega_Q}(\pi) d_\Omega(\omega, \widehat{\omega});$$

*where*

$$L_{\omega_Q}(\pi) = \frac{L_{\omega_r} + \gamma L_{\omega_p} L_{V^\pi}(\omega)}{1 - \gamma}, \qquad L_{V^\pi}(\omega) = \frac{L_r(\omega)(1 + L_\pi)}{1 - \gamma L_P(\omega)(1 + L_\pi)}$$

Given this result, also the return function $j_\omega(\pi)$ is $L_{\omega_Q}$-CLC. In simpler terms, theorem 4.1 exploits the LC property to derive an upper bound on the distance between the $Q_\omega^\pi$ functions of two tasks $\omega_1, \omega_2 \in \Omega$. This result represents an important guarantee on the generalization capabilities of the approach, as it provides a boundary on the error obtained in testing by making inference on a $Q$ function based on the training tasks.

## 5. Fitted Q-Iteration on Meta-MDP

We now define our approach to learn a dynamic step size in the framework of a meta-MDP. As a meta-RL approach, the objectives of our algorithm are to improve the generalization capabilities of PG methods and to remove the need of manually tuning the learning rate for each task. The optimal dynamic step size identification serves two purposes: maximizing the convergence speed and improving the overall training stability, especially when the return is near to the optimum or the current region is uncertain. To accomplish these goals, we propose the adoption of the Fitted Q-Iteration (FQI) (Ernst et al., 2005) algorithm, which is an off-policy, and offline algorithm designed to learn a good approximation of the optimal action-value function by exploiting the Bellman optimality operator. The approach consists in the application of Supervised Learning techniques as, in our case, Extra Trees (Geurts et al., 2006), in order to generalize the $Q$ estimation over the entire state-action space. The algorithm considers a dataset of tuples representing an interaction with the environment, and at each iteration, the regression algorithm can estimate the value function with a planning horizon increased of one step. The FQI version implemented involves Clipped Double Q-learning, introduced by Fujimoto et al. (2019), to penalize the overestimation bias induced by the maximum operator. The approach consists in maintaining two parallel functions $Q_N^{\{1,2\}}$ for each iteration and choosing the action $h$ that maximizes a convex combination of the minimum and the maximum between them, weighted by an external parameter $\lambda > 0.5$:

$$ l + \tilde{\gamma} \max_{h \in \mathcal{H}} \left[ \lambda \min_{j=1,2} Q^j \left( x', h \right) + (1 - \lambda) \max_{j=1,2} Q^j \left( x', h \right) \right]. $$

The dataset generation procedure consists in collecting learning trajectories on the meta-MDP, where the context is sampled as well as the initial policy, while the next policies are computed through iterations of NGA with random step sizes. Once the dataset is generated, the meta-RL version of FQI can be applied, where the goal is to learn the action value function of the Meta-MDP. In this way, for each policy and context, once the natural gradient is estimated, the model can evaluate the best action from a discretization of $\mathcal{H}$.

## 6. Experimental Evaluation

In this section, we show an empirical analysis of the performance of our approach in different environments. When FQI models are trained, the iterations are compared and the best one is shown in figure 1, along with NGA performed with fixed stepsize, evaluated on the same 20 random test tasks and initial policies.

For our first evaluation of the approach, we reproduce Navigation2D, presented in Finn et al. (2017). This environment consists of a unit square space in which an agent aims to reach a random goal in the plane, sampled uniformly in the unit square. In our second experiment, inspired by Penner (2002); Tirinzoni et al. (2019), we consider the scenario of a flat mini golf green, where the context is the putter length and the friction coefficient. Successively, we examine the CartPole balancing task (Barto et al., 1990), that consists in a pole attached to a cart, which the agent has to move to balance the pole as long as possible. The CMDP is induced by varying two environment parameters, the pole mass and length. As a last environment, we considered the Half-cheetah locomotion problem introduced in Finn et al. (2017), where a planar cheetah has to learn to run with a specific goal velocity.

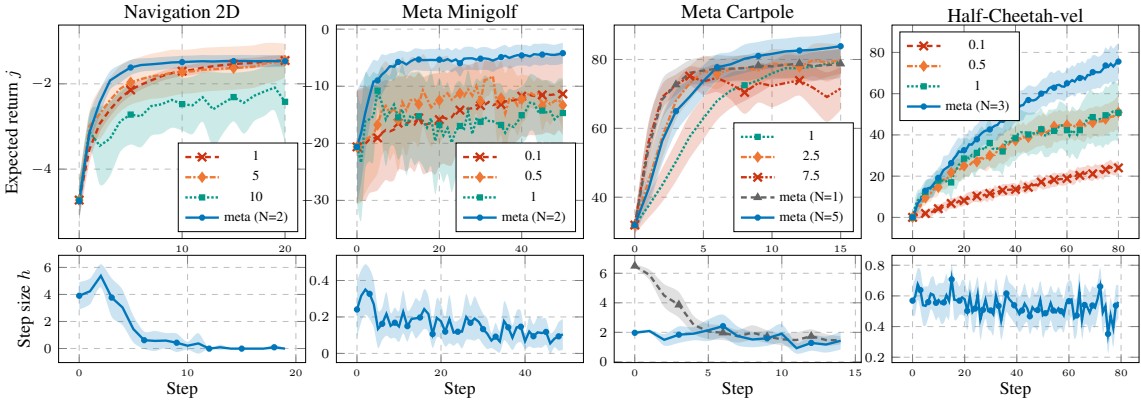

Figure 1: FQI model performance on 20 random test context against fixed step size. The top plots show the 95% c.i. of the expected returns (or the return gain $j(\boldsymbol{\theta}_t) - j(\boldsymbol{\theta}_0)$ for Half-Cheetah environment). The bottom plots show the meta action chosen through learning iterations (95% c.i.). $N$ represents the FQI iteration selected.

As we can note in figure 1, the algorithm is able to select the stepsizes with a good return gain without suffering from any drop. In Navigation and Minigolf environments, the model is able to calibrate its actions, starting with larger steps and slowing down once the policies obtain good results, and all trajectories reach convergence in fewer steps than any other method and consistently reach the optimal values with a low variance. In Meta Cartpole environment, it is possible to see that the best model (blue solid line) is choosing to update the policy with small learning rates: this leads to a lower immediate return gain (high rates have a better learning curve in the first steps), but allows to improve the overall meta return. Finally, we can see the performance gain $j(\boldsymbol{\theta}_t) - j(\boldsymbol{\theta}_0)$ in Half-Cheetah environment. In this case, the FQI model is clearly learning faster than benchmarks, although being far from convergence. The interesting fact is that the fixed learning rates between 0.5 and 1 have almost the same gain return; however, even though the meta actions chosen by the model are almost always within this range, it is still able to adapt the steps to get higher meta-rewards.

## 7. Discussion and Future Work

In this work, we considered the problem of hyperparameter tuning for policy gradient-based algorithms in contextual MDPs, modeling the general problem through the definition of the meta-MDP, for which any policy update rule can be optimized by an agent whose reward is learning. We analyzed the case of Lipschitz meta-MDPs, deriving some general guarantees that hold if the decision process evolves smoothly with respect to the context parametrization. Finally, we implemented the FQI algorithm on the meta-MDP whit natural gradient ascent as update rule, and used it to choose an adaptive stepsize through the learning process. The approach has been evaluated in different settings, where we observed good generalization capabilities of the model, thanks to which it is possible to reach fast convergence speed and robustness without the need of manual fine hyperparameter tuning.

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

## Appendix A. Proofs

In this part of the appendix, we provide the proofs of the results shown in the main paper.

### A.1 Lipschitz continuity of the action-value function

Before describing the proof for Theorem 4.1, we need to recall the Bellman Operator $T^\pi$ for the Action Value Function $Q_\omega^\pi$:

$$T^\pi Q_\omega^\pi(s,a) = R_\omega(s,a) + \gamma \int_{\mathcal{S}} P_\omega(s'|s,a) \int_{\mathcal{A}} Q_\omega^\pi(s',a')\pi(a'|s')dads' \tag{1}$$

$$= R_\omega(s,a) + \gamma \int_{\mathcal{S}} P_\omega(s'|s,a)V_\omega^\pi(s')ds' \tag{2}$$

where $Q_\omega^\pi$ is the fixed point.

Moreover, let's consider a preliminary result on the LC-continuity of the value functions:

**Lemma 1 (Lipschitz value functions)** *Given an $(L_P, L_R)$-LC MDP and a $L_\pi$-LC stationary policy $\pi$, if $\gamma L_P(1 + L_\pi) < 1$, then the Q-function $Q^\pi$ is $L_{Q^\pi}$-LC and the V function is $L_{V^\pi}$-LC w.r.t. the joint state-action space;*

$$L_{Q^\pi} = \frac{L_R}{1 - \gamma L_P(1 + L_\pi)}; \qquad L_{V^\pi} = L_{Q^\pi}(1 + L_\pi) \tag{3}$$

**Theorem 4.1** *Let $\mathcal{M}$ be a $(L_{\omega_P}, L_{\omega_r})$-CLC CMDP for which $\mathcal{M}(\omega)$ is $(L_P(\omega), L_r(\omega))$-LC $\forall \omega \in \Omega$. Given a $L_\pi$-LC policy $\pi$, the action value function $Q_\omega^\pi(s,a)$ is $L_{\omega_Q}$-CLC w.r.t. the context $\omega$, i.e.:*

$$\left| Q_\omega^\pi(s,a) - Q_{\widehat{\omega}}^\pi(s,a) \right| \leq L_{\omega_Q}(\pi)d_\Omega(\omega, \widehat{\omega});$$

*where*

$$L_{\omega_Q}(\pi) = \frac{L_{\omega_r} + \gamma L_{\omega_p} L_{V^\pi}(\omega)}{1 - \gamma}, \qquad L_{V^\pi}(\omega) = \frac{L_r(\omega)(1 + L_\pi)}{1 - \gamma L_P(\omega)(1 + L_\pi)}$$

**Proof** We follow the same ideas as in Rachelson and Lagoudakis (2010a): first of all, given an $L_{\omega_Q}$-LC continuous Q function $Q^\pi$ w.r.t. the task space $\Omega$, the related value function $V_\omega^\pi$ is $L_{\omega_Q}$-LC. Indeed,

$$\left| V_\omega^\pi(s) - V_{\widehat{\omega}}^\pi(s) \right| = \left| \int_{\mathcal{A}} \pi(a \mid s) \left( Q_\omega^\pi(s,a) - Q_{\widehat{\omega}}^\pi(s,a) \right) da \right|$$

$$\leq \int_{\mathcal{A}} \pi(a \mid s) \left| Q_\omega^\pi(s,a) - Q_{\widehat{\omega}}^\pi(s,a) \right| da \tag{4}$$

$$\leq \max_a \left| Q_\omega^\pi(s,a) - Q_{\widehat{\omega}}^\pi(s,a) \right| \leq L_{\omega_Q} d_\Omega(\omega, \widehat{\omega}).$$

Now, we consider the iterative application of Bellman Operators, in such a way that $Q_\omega^{\pi,n+1} = T^\pi Q_\omega^{\pi,n}$, and we prove that $Q_\omega^{\pi,n}$ is $L_{\omega_Q}^n$-LC continuous, and that satisfies the recurrence relation:

$$L_{\omega_Q}^{n+1} = L_{\omega_r} + \gamma L_\pi L_V(\omega) + \gamma L_{\omega_Q}^n. \tag{5}$$

Indeed, for $n = 1$ the property holds immediately, since:

$$\left|Q_\omega^{\pi,1}(s,a) - Q_{\widehat\omega}^{\pi,1}(s,a)\right| = \left|R_\omega(s,a) - R_{\widehat\omega}(s,a)\right| \leq L_{\omega_r} d_\Omega(\omega,\widehat\omega). \tag{6}$$

Now, let us suppose the property holds for $n$. Then:

$$\left|Q_\omega^{\pi,n+1}(s,a) - Q_{\widehat\omega}^{\pi,n+1}(s,a)\right| =$$

$$\left|R_\omega(s,a) - R_{\widehat\omega}(s,a) + \gamma \int_{\mathcal{S}} P_\omega\left(s' \mid s,a\right) V_\omega^{\pi,n}\left(s'\right) ds' - \gamma \int_{\mathcal{S}} P_{\widehat\omega}\left(s' \mid s,a\right) V_{\widehat\omega}^\pi\left(s'\right) ds'\right|$$

$$\leq L_{\omega_r} d_\Omega(\omega,\widehat\omega) + \gamma\left|\int_{\mathcal{S}} \left(P_\omega\left(s' \mid s,a\right) - P_{\widehat\omega}\left(s' \mid s,a\right)\right) V_\omega^{\pi,n}\left(s'\right) ds'\right|$$

$$+\gamma\left|\int_{\mathcal{S}} P_{\widehat\omega}\left(s' \mid s,a\right)\left(V_\omega^{\pi,n}\left(s'\right) - V_{\widehat\omega}^{\pi,n}\left(s'\right)\right) ds'\right|$$

$$\leq L_{\omega_r} d_\Omega(\omega,\widehat\omega) + \gamma L_V(\omega) \sup_{\|f\|_L \leq 1}\left\{\left|\int_{\mathcal{S}} \left(P_\omega\left(s' \mid s,a\right) - P_{\widehat\omega}\left(s' \mid s,a\right)\right) f\left(s'\right) ds'\right|\right\}$$

$$+\gamma \max_{s'}\left|V_\omega^{\pi,n}\left(s'\right) - V_{\widehat\omega}^{\pi,n}\left(s'\right)\right|$$

$$\leq \left(L_{\omega_r} + \gamma L_{\omega_P} L_V(\omega) + \gamma L_{\omega_Q}^n\right) d_\Omega(\omega,\widehat\omega).$$

Now, if the sequence $L_{\omega_Q}^n$ is convergent, it converges to the fixed point of the recurrence equation:

$$L_{\omega_Q} = L_{\omega_r} + \gamma L_{\omega_P} L_V(\omega) + \gamma L_{\omega_Q}. \tag{7}$$

Hence the limit point is the one expressed in 4.1, and the sequence can be proven to be convergent since $\gamma < 1$. ■

As a consequence, the Proof that $j_\omega(\pi)$ is CLC under $\omega$ is immediate:

$$\left|j_\omega(\pi) - j_{\widehat\omega}(\pi)\right| = \left|\int_{\mathcal{S}} \mu\left(s_0\right)\left[V_\omega^\pi\left(s_0\right) - V_{\widehat\omega}^\pi\left(s_0\right)\right] ds_0\right|$$

$$\leq \int_{\mathcal{S}\times\mathcal{A}} \mu\left(s_0\right) \pi\left(a \mid s_0\right)\left|Q_\omega^\pi\left(s_0,a\right) - Q_{\widehat\omega}^\pi\left(s_0,a\right)\right| da\,ds_0$$

$$\leq L_{\omega_Q}(\pi) d_\Omega(\omega,\widehat\omega).$$

## Appendix B. FQI: dataset generation

In this section, we provide a synthetic description for FQI algorithm in the Meta-MDP. The algorithm considers a full dataset $\mathcal{F} = \{(x_t^k, h_t^k, l_t^k, x_{t+1}^k)\}_k$, where each tuple represents an interaction with the meta-MDP: in the $k-$th tuple, $x_t^k$ and $x_{t+1}^k$ are respectively the current and next meta-state, $h_t^k$ the meta-action and $l_t^k$ the meta reward function, as described in Section 3. In order to consider each meta-state $x$, there is the need to sample $n$ trajectories in the inner MDP to estimate return and gradient. At the iteration $N$ of the algorithm, given the (meta) action-value function $Q_{N-1}$, the training set $TS_N = \{(i^k, o^k)\}_k$ is built, where each input is equivalent to the state-action pair $i^k = (x_t^k, h_t^k)$, and the target is the result of

---

**Algorithm 1** Meta-MDP Dataset Generation for NGA (trajectory method)

---

**Input:** CMDP $\mathcal{M}$, distribution over tasks $\psi$, policy space $\Theta$, distribution over initial policies $\rho$, number of meta episodes $K$, number of learning steps $T$, number of inner trajectories $n$.

**Initialize:** $\mathcal{F} = \{\}$,

**for** $k = 1, \ldots, K$ **do**

    Sample context $\omega \sim \psi(\Omega)$ and initial policy $\boldsymbol{\theta}_0 \sim \rho(\Theta)$

    Sample $n$ trajectories in task $\mathcal{M}_\omega$ under policy $\pi(\boldsymbol{\theta}_t)$ and Estimate $j_\omega(\boldsymbol{\theta}_0), \widehat{g}_n(\boldsymbol{\theta}_0, \omega)$

    **for** $t = 0, \ldots, T - 1$ **do**

        Sample meta-action $h \in \mathcal{H}$ and Update policy $\boldsymbol{\theta}_{t+1} = \boldsymbol{\theta}_t + h \frac{\widehat{g}_N(\boldsymbol{\theta}_t, \omega)}{\|\widehat{g}_n(\boldsymbol{\theta}_t, \omega)\|}$

        Sample $n$ trajectories in $(\mathcal{M}_\omega, \pi(\boldsymbol{\theta}_t))$ and estimate $j_\omega(\boldsymbol{\theta}_{t+1}), \widehat{g}_n(\boldsymbol{\theta}_{t+1}, \omega)$

        Set $x = \langle \boldsymbol{\theta}_t, \widehat{g}_n(\boldsymbol{\theta}_t, \omega), \omega \rangle$, $x' = \langle \boldsymbol{\theta}_{t+1}, \widehat{g}_n(\boldsymbol{\theta}_{t+1}, \omega), \omega \rangle$ , $l = j_\omega(\boldsymbol{\theta}_{t+1}) - j_\omega(\boldsymbol{\theta}_t)$.

        Append $\{(x, h, x', l)\}$ to $\mathcal{F}$

    **end for**

**end for**

**Output:** $\mathcal{F}$

---

Bellman optimal operator: $o^k = l_t^k + \gamma \max_{h \in \mathcal{H}} Q_{N-1}(x_{t+1}^k, h)$. In this way, the regression algorithm adopted is trained on $TS$ to learn $Q_N$ with the learning horizon increased of one step. However, as the regression procedures are iterated, new estimation errors are introduced that might cumulate over time, resulting in a degradation of the performances with $N$.

In general, the dataset is created by following $K$ learning trajectories over the CMDP: at the beginning of each meta-episode, a new context $\omega$ and initial policy $\boldsymbol{\theta}_0$ are sampled from $\psi$ and $\rho$; then, for each of the $T$ learning steps, the meta action $h$ is randomly sampled to perform the policy update. In this way, the overall dataset is composed of $KT$ tuples. However, if the policy space is small enough, it is possible to explore the overall task-policy space $\Omega \times \Theta$ through a generative approach: instead of following the learning trajectories, both $\omega, \boldsymbol{\theta}_0$ and $h$ are sampled every time. We refer to this method as "generative" approach, while the former will be denoted as "trajectory" approach.

The pseudo code for the dataset generation process is provided in algorithm 1.

## Appendix C. Experiment Details

In this section, we provide more details regarding the experimental campaign provided. In the following environments, all the policies considered are Gaussian, and linear w.r.t. the state observed (with bias $\theta_0$), i.e. $\pi_{\boldsymbol{\theta}}(a|s) \sim \mathcal{N}(\theta_0 + \boldsymbol{\theta}^\top s, \sigma^2)$, where $\sigma$ is fixed standard deviation, with a different setting for each environment.

### C.1 Navigation2D Description

The Navigation2D environment consists of a 2-dimensional square space in which an agent, represented as a point, aims to reach a goal in the plane traversing the minimum distance.

At the start of the episode, the agent is placed in the initial position $s_0 = (0, 0)$ of the Cartesian plane. Then, at each step $t$ the agent observes its current position and performs

an action $a_t$ corresponding to movement speeds along the $x$ and $y$ axes:

$$a_t = (v_x, v_y), \text{ where } v_x, v_y \in [-v^{\max}, v^{\max}]. \tag{8}$$

According to this action, the agent can move in every direction of the plane, with a limit on the maximum speed $v^{\max} = 0.1$ allowed in a single step. This parameters determines the minimum number of steps necessary to reach the goal and can be varied to tune the difficulty of the environment.

At each step, the environment produces a reward equal to the negative Euclidean distance from the goal:

$$r_t = \sqrt{(x_t - x_{\text{goal}})^2 + (y_t - y_{\text{goal}})^2}. \tag{9}$$

An episode terminates when the agent is within a threshold distance $d_{\text{thresh}}$ from the goal or when the horizon $H = 10$ is reached.

The distribution of tasks is implemented as a CMDP $\mathcal{M}(\omega)$ in which, at each episode, a different goal point is selected at random. The context $\omega$ is given by a 2D vector, such that:

$$\omega = (x_{\text{goal}}, y_{\text{goal}}), \text{ where } x_{\text{goal}}, y_{\text{goal}} \sim U(-1, 1). \tag{10}$$

Parameters used for experiments:

- initial policy distribution $\rho = \mathcal{N}(0, 0.1)$;
- discount factor $\gamma = 0.99$;
- policy standard deviation $\sigma = 1.001$;
- task distribution $\psi = \mathcal{U}([-0.5, 0.5]^2)$;
- FQI dataset method: trajectories;
- FQI number of samples: $K = 4000$ with learning horizon $T = 20$;
- inner trajectories $n = 200$ with horizon $H = 10$;
- number of estimators $= 50$, minimum samples split $= 0.01$;
- step size $\mathcal{H} = [0, 8]$;
- step size sampling distribution: uniform in $\mathcal{H}$;
- step size selected in evaluation from an evenly spaced discretization of 101 values in $\mathcal{H}$.

### C.2 Minigolf Description

In the minigolf game, the agent has to shoot a ball with radius $r$ inside a hole of diameter $D$ with the smallest number of strokes. The friction imposed by the green surface is modeled by a constant deceleration $d = \frac{5}{7}\rho g$, where $\rho$ is the dynamic friction coefficient between the ball and the ground and $g$ is the gravitational acceleration. Given the distance $x$ of the ball from the hole, the agent must choose the force $a$, from which the velocity of the ball $v$ of the ball is determined as $v = al^2(1 + \epsilon)$, where $\epsilon \sim \mathcal{N}(0, 0.25)$ and $l$ is the putter length. For each distance $x$, the ball falls in the hole if its velocity $v$ ranges from $v_{min} = \sqrt{2dx}$ to $v_{max} = \sqrt{(2D - r)^2 \frac{g}{2r} + v_{min}^2}$. In this case, the episode ends with a reward 0; if $v > v_{max}$ the ball falls outside the green, and the episode ends with a reward -100. Otherwise, if

$v < v_{min}$, the agents gets a reward equal to -1, and the episode goes on from a new position $x_{new} = x_{old} - \frac{v^2}{2d}$. At the beginning of each episode, the initial position is selected from an uniform distribution between 0m and 20m from the hole. The stochasticity of the action implies that the stronger is the action chosen the more uncertain is the outcome, as the effect of r.v. $\epsilon$ become more effective. As a result, when it is away from the hole, the agent might not prefer to try to make a hole in one shot, preferring to perform a sequence of closer shots. In this case, the context is given by the friction coefficient $\rho \in [0.065, 0.196]$ and by the putter length $l \in [0.7, 1]m$.

During the experiment, the environment parameters are set to imitate the dynamics of a realistic shot in a minigolf green, within the limits of our simplified simulation. This is the complete configuration adopted:

- horizon $H = 20$;
- discount factor $\gamma = 0.99$;
- angular velocity $\omega \in [1 \times 10^{-5}, 10]$;
- initial distance $x_0 \in [0, 20]$ meters;
- ball radius $r = 0.02135$ meters;
- hole diameter $D = 0.10$ meters;

The distribution of tasks is built as a CMDP $\mathcal{M}(\omega)$, induced by the pair $\omega = (l, \rho)$. At each meta episode, a new task is sampled from a multivariate uniform distribution within this ranges:

- putter length $l \sim U(0.7, 1)$ meters;
- friction coefficient $\rho \sim U(0.065, 0.196)$.

Parameters used for experiments:

- initial policy distribution $\boldsymbol{\theta} = (w, b) \sim U((-1, 2), (-2, 3.5))$ (2-dimensional policy);
- policy standard deviation $\sigma = 0.1$;
- FQI dataset method: generative;
- FQI number of samples: $K = 10000$;
- inner trajectories $n = 400$ with horizon $H = 20$;
- number of estimators $= 50$, minimum samples split $= 0.01$;
- step size space: $\mathcal{H} = [0, 1]$
- step size sampling distribution: uniform in $\mathcal{H}$;
- step size selected in evaluation from an evenly spaced discretization of 101 values in $\mathcal{H}$.

### C.3 CartPole description

The CartPole environment (Barto et al., 1990), also known as the Inverted Pendulum problem, consists in a pole attached to a cart by a non actuated joint, making it an inherently unstable

system. The cart can move horizontally along a frictionless track to balance the pole. The objective is to maintain the equilibrium as long as possible.

In this implementation, an episode starts with the pendulum in vertical position. At each step, the agent observes the following 4-tuple of continuous values:

- cart position $x_{\text{cart}} \in [-4.8, 4.8]$;
- cart velocity $v_{\text{cart}} \in \mathbb{R}$;
- pole angle $\phi_{\text{pole}} \in [-0.418, 0.418]$ rad;
- pole angular velocity $\omega_{\text{pole}} \in \mathbb{R}$.

Given the state, the agent chooses an action between 0 and 1 to push the cart to the left or to the right. For each step in which the pole is in balance, the environment produces a reward of $+1$. An episode ends when the pole angle from the vertical position is higher than 12 degrees, or the cart moves more than 2.4 units from the center, or the horizon $H = 100$ is reached.

In our experiments, some parameters are fixed, as the mass of the cart ($m_{cart} = 1$ kg), the length of the pole ($l_{pole} = 0.5$ m) and the force applied by the cart ($F = 10$ N). The CMDP $\mathcal{M}(\omega)$ is induced by varying two environment parameters, the pole mass $m_{pole}$ and the pole length $l_{pole}$, that form the context parameterization $\omega = (m_{pole}, l_{pole})$. Each task in the meta-MDP is built by sampling $\omega$ from a multivariate uniform distribution, within these ranges:

- pole length $l_{pole} \sim U(0.5, 1.5)$m;
- pole mass $m_{pole} \sim U(0.1, 2)$ kg.

Parameters used for experiments:

- initial policy distribution $\boldsymbol{\theta_d} \sim \mathcal{N}(0, 0.1)$ for each component $\boldsymbol{\theta}_d$;
- policy standard deviation $\sigma = 1.001$;
- meta-discount factor $\widetilde{\gamma} = 1$;
- FQI dataset method: trajectories;
- FQI number of samples: $K = 3200$ with learning horizon $T = 15$;
- inner trajectories $n = 100$ with horizon $H = 100$;
- number of estimators $= 150$, minimum samples split $= 0.05$;
- step size $\mathcal{H} = [0, 10]$;
- step size sampling distribution: uniform in $\mathcal{H}$;
- step size selected in evaluation from an evenly spaced discretization of 101 values in $\mathcal{H}$.

## C.4 Half Cheetah description

The CMDP $\mathcal{M}(\omega)$ is induced by varying the goal velocity of the half cheetah $v_{goal}$, which defines the context $\omega$, with uniform distribution $U(0, 2)$.

Parameters used for experiments:

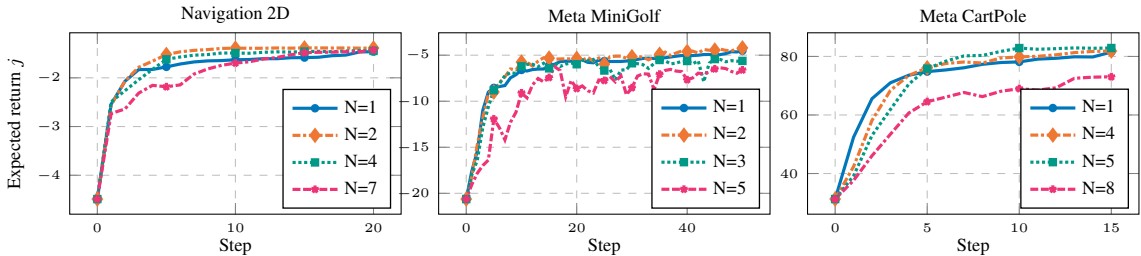

Figure 2: FQI model performance among different iterations. For the sake of clarity, only the average values are shown.

- initial policy distribution $\boldsymbol{\theta_d} \sim \mathcal{N}(0, 0.1)$ for each component $\boldsymbol{\theta}_d$;
- policy standard deviation $\sigma = 1.001$;
- meta-discount factor $\widetilde{\gamma} = 1$;
- FQI dataset method: trajectories;
- FQI number of samples: $K = 200$ with learning horizon $T = 80$;
- inner trajectories $n = 100$ with horizon $H = 100$;
- number of estimators $= 150$, minimum samples split $= 0.05$;
- step size $\mathcal{H} = [0, 1]$;
- step size sampling distribution: uniform in $\mathcal{H}$;
- step size selected in evaluation from an evenly spaced discretization of 101 values in $\mathcal{H}$.

## Appendix D. Other results

### Comparison among FQI Iterations.

As said, as the regression procedures are iterated in the application of FQI algorithm, there is a trade-off between a larger planning horizon and the accumulation of new regression errors. In figure 2 we show some of the learning curves with different FQI iterations. For all the environments considered, it is possible to see that the direct regression on the meta reward (i.e. one FQI iteration) does not provide the best performances, while from a certain point the results start to get worse. As far as Meta Cartpole environment is concerned, we can clearly see that the models select progressively more cautious steps in order to improve learning, as explained in section 6.

### Comparison with learning rate schedules.

In figure 1 we compared the FQI models with the choice of a fixed step size. Other schedules are often considered, as for example a dynamic step size $h_{t+1} = \frac{\alpha}{t}$ (or, similarly, an exponentially decreasing learning rate $h_{t+1} = \alpha h_t$). In figure 3, the baselines considered are three different starting step sizes $\alpha$, which are compared to our approach, which is still proving the best performance.

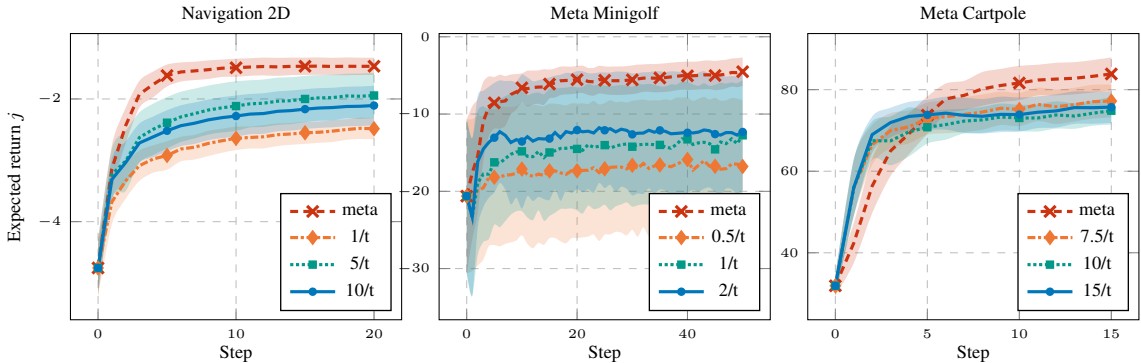

Figure 3: FQI model performance on 20 random test context against a decreasing step size (95% c.i.).

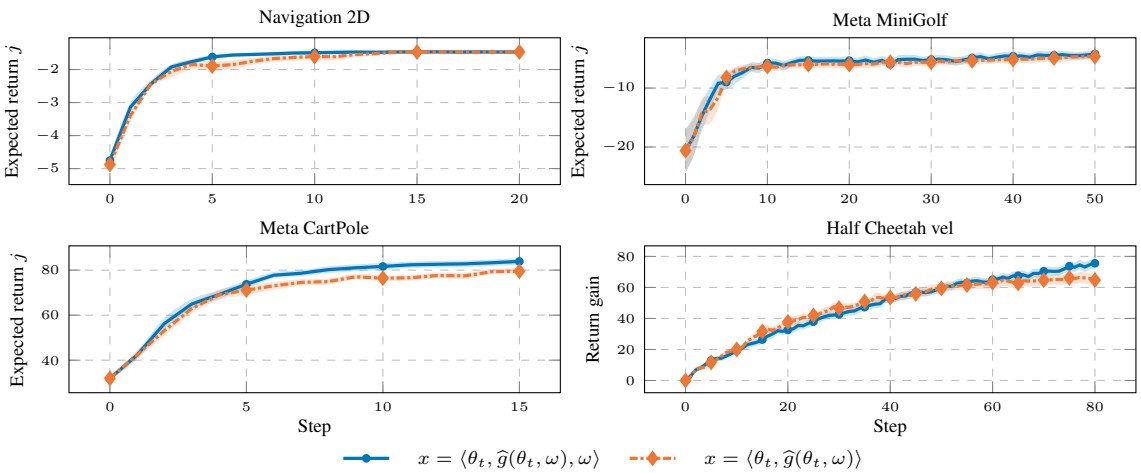

Figure 4: FQI model performance obtained by considering or excluding the explicit task parametrization $\omega$. (95% c.i.)

## Explicit knowledge of the context: is it informative?

In the experimental campaign, we assumed to be able to represent the parametrized context $\omega$, as this information can be used to achieve an *implicit task-identification* by the agent. However, in some cases the external variables influencing the process might be not observable. However, the gradient itself already implicitly includes information regarding the transition and reward probabilities: what is lost when we do not consider the explicit parametrization of the task? In general, as we can see in figure 4, there is no big loss in the performance, especially for Minigolf environment; however, in Meta CartPole, the task parametrization seem to be informative to the choice of the step size.

