# OpenReview forum: "Meta Learning the Step Size in Policy Gradient Methods"
_ICML.cc/2021/Workshop/AutoML — AutoML@ICML2021 Poster_

### Official Review · Reviewer_SD7H · 2021-06-04
**Interesting framework, experiments are a work in progress...**

**Rating:** 6
**Confidence:** 3

**Review:**

SUMMARY

This paper provides an approach to learn learning rate schedules for RL tasks. In doing so, the authors introduce a "Meta-MDP" as a family of MDPs with shared structure. They then propose to use Fitted Q-Iteration to learn a learning rate schedule. The authors provide some theoretical results for the case where the MDPs are Lipschitz.

The experiments in this paper are extremely weak - not only are they toy problems but they are not statistically significant, and clearly overfit to the best value of N (see Fig2). However, I am voting for a borderline *accept* solely because it is an interesting topic and a framework which others at the workshop could find interesting. For a journal/conference the experiments need to be significantly revisited (see below).

POSITIVES

-It is interesting to see an attempt to provide a framework for learning hyperparameters for novel RL tasks. This type of approach has been shown to be empirically successful in a variety of settings, e.g. using bandits. However I believe this is the first approach to set up the problem in this way.

-The theoretical results are some of the first such results for this type of problem. It may lead to future work.

NEGATIVES

-The gains in some experiments vs. a fixed learning rate are trivially small (especially Navigation and Cartpole). Despite running with many seeds, there is no statistical significance to the gain. Given that the experiments are toy, it would be hoped we could at least have a solid conclusion from them, which there is not.

-It is clear from the Appendix (Fig 2) that the results were cherry picked. For Cartpole, N=1,4,7,9 all perform worse than fixed learning rates, but the authors present N=6 which outperforms. Thus - they introduce a new hyperparameter which may be even more sensitive than the one they are tuning!

-It is unclear why we would do this over existing methods like meta-gradients (or HOOF), which the authors cite in the intro but never compare against or discuss.

-The evaluation is done in-distribution, thus assuming we have access to the full distribution of tasks, possibly even training on the test task at some point during training (this doesn't seem to be specified anywhere).

-It is not clear when the Lipschitz assumptions hold, and this is not discussed.

---

### Official Review · Reviewer_yZHV · 2021-06-13
**Exiting paper with promising results**

**Rating:** 8
**Confidence:** 4

**Review:**

The paper formalizes meta learning as a sequential decision making problem, and for the space of step size hyperparameter shows Lipshitz continuity. From those theoretical foundations, the paper goes to solve step size selection for several meta learning tasks.

It's refreshing to see a formal definition of meta-MDP and the analysis around continuity. This is the strongest aspect of the paper, and the direction that ought to be pursued further. The step size has received a relatively little converge, despite its importance.

It is also great to see the evaluation on several tasks along with the baselines.

The biggest downside to this paper is that the Appendix does a lot of the heavy lifting. The transition from the meta-MDP to the evaluation is pretty sudden without the implementation details. This is more of the cosmetic issue, given the paper size constraints. Clearly this is a summary of a larger body of work.

Note to authors -- please update the template and use AutoML.

---

### Decision · Program_Chairs · 2021-06-21

Accept (Poster)